# QTL Mapping and Transcriptome Analysis Reveal Candidate Genes Regulating Seed Color in *Brassica napus*

**DOI:** 10.3390/ijms24119262

**Published:** 2023-05-25

**Authors:** Fangying Liu, Hao Chen, Liu Yang, Liang You, Jianye Ju, Shujie Yang, Xiaolin Wang, Zhongsong Liu

**Affiliations:** College of Agronomy, Hunan Agricultural University, Changsha 410128, China

**Keywords:** rapeseed, yellow seed, genetic map, quantitative trait locus, coexpression network, flavonoid

## Abstract

Yellow seeds are desirable in rapeseed breeding because of their higher oil content and better nutritional quality than black seeds. However, the underlying genes and formation mechanism of yellow seeds remain unclear. Here, a novel yellow-seeded rapeseed line (Huangaizao, HAZ) was crossed with a black-seeded rapeseed line (Zhongshuang11, ZS11) to construct a mapping population of 196 F_2_ individuals, based on which, a high-density genetic linkage map was constructed. This map, comprising 4174 bin markers, was 1618.33 cM in length and had an average distance of 0.39 cM between its adjacent markers. To assess the seed color of the F_2_ population, three methods (imaging, spectrophotometry, and visual scoring) were used and a common major quantitative trait locus (QTL) on chromosome A09, explaining 10.91–21.83% of the phenotypic variance, was detected. Another minor QTL, accounting for 6.19–6.69% of the phenotypic variance, was detected on chromosome C03, only by means of imaging and spectrophotometry. Furthermore, a dynamic analysis of the differential expressions between the parental lines showed that flavonoid biosynthesis-related genes were down-regulated in the yellow seed coats at 25 and 35 days after flowering. A coexpression network between the differentially expressed genes identified 17 candidate genes for the QTL intervals, including a flavonoid structure gene, *novel4557* (*BnaC03.TT4*), and two transcription factor genes, namely, *BnaA09G0616800ZS* (*BnaA09.NFYA8*) and *BnaC03G0060200ZS* (*BnaC03.NAC083*), that may regulate flavonoid biosynthesis. Our study lays a foundation for further identifying the genes responsible for and understanding the regulatory mechanism of yellow seed formation in *Brassica napus*.

## 1. Introduction

*Brassica napus* (AACC, 2*n* = 38), also known as rapeseed or canola, originated in Europe and was formed by an interspecific hybridization between *B. rapa* (AA, 2*n* = 20) and *B. oleracea* (CC, 2*n* = 18), with subsequent spontaneous chromosome doubling, 7500 years ago [1]. It is one of the major oilseed crop species across the world and provides vegetable oil and biofuels for humans, as well as stock feed for animals. Compared to black seeds, yellow seeds are preferred for their higher oil content due to their thinner seed coat and lower fiber, lignin, and pigment contents [2,3]. Although there is a lack of natural yellow-seeded resources in *B. napus*, researchers have developed yellow-seeded *B. napus* lines through distant hybridization [4,5]. In our previous study, a genetically stable yellow-seeded *B. napus* inbred line, named HAZ, was developed from an interspecific cross of *B. juncea* × *B. napus* [6].

The inheritance of seed color in rapeseed is complex. Seed color is mainly controlled by the maternal genotype, but a pollen effect and embryonic control have also been reported [7,8], and environmental factors, such as temperature and red and blue light, also play roles in seed coloration [9,10]. Since F_1_ plants produce yellow seeds or black seeds under different genetic backgrounds, the inheritance of the yellow-seed trait is considered to follow both partially dominant and recessive genetic models [11]. The seed color in *B. napus* F_2_ populations shows continuous variation and three to four gene loci are reportedly involved in the determination of the seed color [7,11]. Quantitative trait locus (QTL) mapping has revealed a few major QTLs on chromosomes A09 or C08 [7,11,12] and stable yellow-seeded mutants have also been generated via targeted mutations of *BnTT8* or *BnaTT2* [13,14]. However, the genes responsible for the seed color formation in *B. napus* and their regulation mechanism remain unclear.

In yellow seeds, owing to their transparent and colorless testa, the yellow color of their embryos is visible from the outside, while black/brown seeds show a black/brown color in their testa. Proanthocyanidins (PAs) are regarded as the main components contributing to the seed color in *Brassica* species [15,16]; these compounds are synthesized via the flavonoid biosynthesis pathway and accumulate specifically in the endothelial layer of the seed coat [17]. In *Arabidopsis thaliana*, the flavonoid biosynthesis pathway is quite clear [15,18]. This pathway occurs at the convergence of the phenylpropanoid and acetate pathways. In the phenylpropanoid pathway, under the catalysis of phenylalanine ammonia-lyase (PAL), cinnamic acid 4-hydrolase (C4H), and p-coumaric acid:CoA ligase (4CL), phenylalanine is converted into p-coumaroyl-CoA, which, together with malonyl-CoA provided by the acetate pathway, serves as a substrate for flavonoid biosynthesis. Flavonoids are then synthesized by the early biosynthetic genes (EBGs) (*TT4*, *TT5*, *TT6*, and *TT7*) and late biosynthetic genes (LBGs) (*TT3*, *TT18*, and *BAN*) in the cytoplasm. Other genes *(TT9*, *TT12*, *TT13,* and *TT19*) are responsible for the transport of flavonoids into vacuoles. Finally, *TT10* is involved in the oxidative polymerization of flavonoids and the formation of insoluble brown PAs. EBGs are common flavonoid pathway genes that are positively regulated by a class of R2R3–MYB transcription factors (MYB11/MYB111/MYB112) [19], while LBGs are involved in the biosynthesis of anthocyanins and PAs and are positively regulated by MYB–bHLH–WD40 (MBW) ternary complexes. In seeds, the MWB (TT2-TT8-TTG1) complex is the main PA biosynthesis regulator [20]. In addition, flavonoid biosynthesis is also positively regulated by *TT1*, *TTG2*, *TT16*, *MYB113*, *MYB114*, *TCP3*, *HY5,* and *NAC2* and negatively regulated by *MYBL2*, *MYB4*, *MYB7*, *MYB32*, *CPC*, *SPL9*, *STK*, *MIR828*, *MIR858*, and members of the *LBD* family [21,22,23,24,25,26].

High-density genetic maps are crucial for high-resolution QTL mapping. In contrast to single-nucleotide polymorphism (SNP) arrays and reduced representation sequencing, whole-genome resequencing (WGR) provides the most comprehensive genetic variants, thus outperforming other methods in terms of the development of markers, genotyping, and increasing the marker density of genetic maps [27]. Recombinant bin maps, constructed via a sliding window approach based on the genotypes of mapping populations, have been used to identify the important QTLs for the important agronomic traits of crop species [28,29]. A weighted gene coexpression network analysis (WGCNA) classifies the genes that may share biological functions into coexpression modules and has been proven to facilitate the mining of causal genes within the QTL intervals of target traits [30,31]. In this study, a high-density genetic linkage map was constructed via a resequencing of the F_2_ population derived from ZS11 × HAZ, in order to identify the QTLs associated with the seed color in *B. napus*. A coexpression network between the differentially expressed genes (DEGs) located within the QTL intervals and the DEGs associated with flavonoid biosynthesis was constructed to mine the candidate genes. Our results lay a foundation for the cloning of the genes underlying the yellow seed formation and provide genetic resources and theoretical support for the breeding of yellow-seeded varieties in *B. napus*.

## 2. Results

### 2.1. High-Density Bin Map Construction

We generated an F_2_ population derived from a cross between ZS11 and HAZ. In total, 196 F_2_ individuals, as well as their parental lines, were subjected to WGR on an Illumina HiSeq platform. More than 1.4 Tb of filtered data (~4.8 billion reads), with an average depth of 4.8× per individual, were generated (Appendix A). The clean reads were mapped against the ZS11 genome sequence and 400,744 alleles were saved after being filtered, namely, 353,831 SNPs and 46,913 insertions–deletions (InDels), which were used for the genotyping in the F_2_ population. Using a slide window method, we generated 4148 bin marks based on the analysis of the recombination breakpoints. The bin markers were ordered and clustered into 19 groups by MSTmap and the genetic distances between these markers were calculated using the Kosambi function. Finally, a 1618.33 cM high-density genetic map was constructed with an average distance of 0.39 cM between its adjacent markers (Table 1 and Figure 1A). Overall, the genetic map showed a good marker collinearity with the ZS11 genome, with a few exceptions (Figure 1B).

### 2.2. Phenotypic Variation and QTLs Detected for Seed Color

The seed color of HAZ was distinctly different from that of ZS11. The seed coats were subjected to vanillin staining at 15, 25, and 35 days after flowering (daf). The seed coats of ZS11 started to turn red at 25 daf and the color became darker at 35 daf, while no seed coats of HAZ turned red at any stage, except for a stained hilum, suggesting that yellow seed coats contain less PAs than black seed coats (Figure 2A). In the F_2_ population, the seed color showed continuous variation, ranging from black, dark brown, and light brown to brown yellow, yellow brown, and yellow (Figure 2B). To quantify the seed color of the F_2_ population, we used both imaging and spectrophotometry. Three color values (R, G, and B) were measured using imaging and other three color values (l, a, and b) were measured using spectrophotometry (Appendix A). All the color values were statistically highly correlated (Appendix A). According to the frequency distribution of the seed color value (R) in the F_2_ generation (Figure 2C), the seed color value (R) increased as the color became lighter, the values of which were 42.95 for ZS11 and 151.24 for HAZ, and in the F_2_ population, they ranged from 41.99 to 138.68. Two peaks and one valley at approximately 67 appeared in the frequency distribution, splitting the F_2_ population into two groups. Interestingly, the group with a seed color value (R) smaller than 67 included plants producing black, dark brown, or light brown seeds, while the other group, with a seed color value (R) larger than 67, included plants producing brown yellow, yellow brown, or yellow seeds. As a result, we separated the F_2_ plants into two grades using visual scoring according to whether yellow seeds were produced. Finally, seven color values were used to conduct the QTL mapping. A common major QTL on chromosome A09, which explained 10.91–21.83% of the phenotypic variance of all the seed color values, was detected. Another QTL on chromosome C03, explaining 6.19–6.69% of the phenotypic variance of the seed color values (R, B, and b), was detected (Table 2, Figure 2D and Appendix A).

### 2.3. Differential Expression Analysis between Yellow and Black Seed Coats

To study the gene expression changes that may induce seed color variation, RNA sequencing (RNA-seq) of the seed coats of the mapping population parents (ZS11 and HAZ) was performed at three developmental stages (15, 25, and 35 daf). After quality control processes were applied, ~6 Gb data were obtained per library and 89.79%–93.34% of the reads were uniquely mapped to the ZS11 genome (Appendix A). Using these RNA reads, we assembled 5842 novel genes and combined them with the annotated genes in the ZS11 genome for a subsequent analysis. A total of 83,088 genes were expressed during the seed coat development. The Pearson correlation coefficients among all three biological replicates from the same material were above 0.96 (Appendix A), showing the high credibility of the data used in this study. The transcript abundances were compared between the parents at the same stage, with ZS11 used as a control. A total of 18,575 DEGs were identified, 4179 of which were common across all stages (Figure 3A). There were far fewer up-regulated genes than down-regulated genes at each stage, namely, 3519:5857, 4183:6060, and 4792:6741 at 15, 25, and 35 daf, respectively (Figure 3B). To clarify the pathways in which the DEGs participated, all the genes were annotated via KOBAS 3.0 and 20,511 genes were assigned to pathways. The DEGs at three stages were subjected to a Kyoto Encyclopedia of Genes and Genomes (KEGG) enrichment analysis, respectively. The results showed that flavonoid biosynthesis was most significantly enriched at 25 and 35 daf, but not at 15 daf (Figure 3C), suggesting that flavonoids were synthesized after 15 daf, which is consistent with the results of the vanillin staining (Figure 2A). Hence, we considered 25 to 35 daf to be an important stage for seed coloration.

### 2.4. Expression Profiles of Genes Involved in Flavonoid Biosynthesis

Since PAs synthesized through the flavonoid biosynthesis pathway constitute a major component of seed color, we investigated the expression levels of the flavonoid-related genes. Fifty-six flavonoid-related genes in *A. thaliana* were used to identify 218 homologs in the ZS11 genome (Appendix A). Except for one copy of *F3H* located on Bnascaffold0027, the other homologs were anchored to all 19 chromosomes (Appendix A). The expression levels of the 31 flavonoid-related genes were extremely low (fragments per kilobase of transcript per million mapped reads (FPKM) < 1 across three replicates) or undetected. Of the other 187 expressed genes, 96 genes showed differential expressions between their parents (Figure 4, Appendix A). The expression of the flavonoid-related genes significantly changed after 15 daf, with only 31 DEGs at 15 daf, but 73 and 70 DEGs at 25 and 35 daf, respectively. The up-regulated DEGs are far less than the down-regulated DEGs, with numbers of 7:66 at 25 daf and 9:61 at 35 daf. In the general phenylpropanoid pathway, three copies of *PAL1*, two copies of *PAL4*, and one copy of *PAL2* were significantly down-regulated at 25 daf. The down-regulation of six *C4H* homologs was found at 25 or 35 daf. Among the *4CL* homologs, both copies of *4CL3*, which may be important for flavonoid biosynthesis, were down-regulated at 25 and 35 daf. For the DEGs related to flavonoid biosynthesis, except for one copy of *TT4* and two negative regulators (*MYB32* and *STK*) that were up-regulated, the other copies of structural genes (*TT4*, *TT5*, *TT6*, *TT7*, *TT3*, *TT18*, *BAN*, and *TT10*), regulatory genes (*TT2*, *TT8*, *TT1*, *TTG2*, and *TT16*), and transport-related genes (*TT12*, *TT13*, and *TT19*) were down-regulated at 25 or 35 daf. These results indicated that the down-regulation of flavonoid-related genes at 25 or 35 daf was an important reason for yellow seed formation. To validate the reliability of the transcriptome analysis, we analyzed the expression levels of 20 DEGs identified in the RNA-seq analysis using qRT-PCR, including 17 DEGs involved in flavonoid biosynthesis. A linear regression analysis showed very high correlation coefficients (R = 0.815–0.880) between the qRT-PCR and RNA-seq analyses at all three developmental stages (Appendix A, Appendix A), confirming the credibility of the RNA-seq analysis.

### 2.5. Gene Coexpression Network Revealed Gene Modules Related to Seed Color

To identify the modules involved in seed coloration, a coexpression network comprising 83,088 expressed genes was constructed, followed by decomposition of the network into 37 coexpressed modules. Each module contained a set of genes whose expressions were significantly correlated with each other and may share biological functions. The largest module, turquoise, contained 21,427 genes, whereas the smallest module, darkmagenta, contained only 114 genes (Figure 5A). Eigenvalues were calculated to represent the overall expression levels of the genes in each module. For each module, the correlations between the eigenvalue and samples were computed. Interestingly, the brown module was positively correlated with ZS11 and negatively correlated with HAZ at all three stages. The darkgrey module exhibited a significantly positive correlation (r^2^ = 0.98 ***) with ZS11 at 25 daf (Figure 5B). The KEGG enrichment showed that the flavonoid biosynthesis pathway was enriched in both the brown and darkgrey modules (Figure 5C). Therefore, these two modules were regarded as key modules for seed color and may harbor major regulators influencing seed color formation.

### 2.6. Candidate Gene Prediction for Seed Color

There were 183 and 710 genes within the intervals of the QTLs on chromosomes A09 and C03, respectively. Among those genes, the expression of 91 genes differed between ZS11 and HAZ at 25 or 35 daf, but only *novel4557* (*BnaC03.TT4*) was included in the identified flavonoid-related genes (Figure 4, Appendix A). *TT4* encodes chalcone synthase (CHS), the key enzyme that catalyzes the first step of flavonoid biosynthesis. In soybean, silencing the expression of CHS with RNA interference inhibited the seed coat pigmentation [32]. In this study, *novel4557* was significantly down-regulated in HAZ at both 25 and 35 daf and belonged to the darkgrey module (Figure 4, Appendix A); hence, this gene was considered to be a valuable candidate. To further screen for candidate genes, a coexpression network between the DEGs within the QTL intervals and the DEGs related to flavonoid biosynthesis was constructed. Sixteen genes within the QTL intervals showed coexpressions with the flavonoid-related DEGs and were also selected as candidates. (Figure 6, Appendix A). According to the network, *BnaA09G0616800ZS* was coexpressed with 21 flavonoid-related DEGs and showed a significant down-regulation in HAZ at 25 and 35 daf. This gene encodes a NF-YA8 transcription factor that activates the transcription of *MIR156* through direct binding to CCAAT cis-elements in their promoters [33]. Since *MIR156* is able to finely regulate the anthocyanin biosynthetic pathway via microRNAs, transcription factors, and structural genes [34], we selected *BnaA09G0616800ZS* as an important candidate gene. In our study, *BnaC03G0060200ZS* was coexpressed with nine flavonoid-related DEGs and down-regulated at 25 daf. *BnaC03G0060200ZS* encodes an NAC transcription factor that plays a role in anthocyanin accumulation [35]. Therefore, we also selected *BnaC03G0060200ZS* as an important candidate.

## 3. Discussion

The number of molecular markers and population size are major factors for determining the resolution of QTL mapping. WGR serves as an effective way of detecting the numerous markers in populations for a genetic map construction. In this study, a high-density genetic map with 4174 recombinant bins was generated; this map covered 1618.33 cM and had an average marker interval of 0.39 cM, which lays a good foundation for subsequent QTL mapping. Compared to maps constructed in previous studies, our genetic map had a comparable density of markers but was shorter [36,37]. Regions in which recombination was suppressed were found on chromosomes C09, C01, A09, and C03 (Figure 1B); these regions had narrow genetic distances but physical distances longer than 10 Mb, which might have been caused by structure variation or methylation [38,39].

In this study, the seed color of the F_1_ plant was yellow brown (Figure 2B). This observation indicated that yellow seeds were partially dominant over black seeds, similar to the findings in previous studies [40,41,42]. Imaging, spectrophotometry, and visual scoring were used to measure the seed color of F_2_ plants. The visual scoring classified the plants into two groups based on whether they produced yellow seeds or not; thus, this did not take the seed color variation within the groups into consideration. This method detected the same major QTLs on chromosome A09 as the imaging and spectrophotometry did. This common QTL colocalized with the major seed color QTL that was also been associated with seed fiber and oil content in previous studies [12,43]. However, the QTL on chromosome C03 could not be detected with the visual scoring. Therefore, we supposed that the QTL on chromosome A09 was the main determinant of the yellow seed formation and had epistatic effects on genes that lead to the production of black seeds, whereas the minor QTL on chromosome C03 was responsible for the proportion of the yellow seeds within the groups.

PAs are one of the end products of the flavonoid biosynthesis pathway and are thought to determine the seed color formation in *B. napus* [17,44]. Vanillin staining, a KEGG enrichment analysis, and flavonoid-related gene expression profiling suggested that the down-regulation of flavonoid-related genes at 25 and 35 daf hindered the flavonoid biosynthesis in the yellow seeds. In *B. rapa* and *B. juncea*, the cloned yellow-seed genes, including *TT8* [45,46], *TT1* [47], and *TTG1* [48,49], are related to PA biosynthesis. However, only *novel4557* (*BnaC03.TT4*) in the QTL region on chromosome C03 is reportedly related to PA biosynthesis. The nearest flavonoid-related gene to the QTL on chromosome A09 was a copy of *TT6* (*BnaA09G0576300ZS*), but this gene was not expressed during any of the three studied seed coat developmental stages in either parent. Differential expression analyses and interaction network constructions (protein-protein interaction networks or coexpression networks) are useful methods for selecting the candidate genes at QTLs. In previous studies, integrating sequence variation annotations, expression differences, and protein–protein interaction networks, *BnaA09.GH3.3*, *BnaA09.JAZ1*, and *BnaA09.LOX3* were selected as the candidate genes for seed color formation in *B. napus* N53-2 [12]. On the basis of a transcriptomic analysis and transcription factor predictions, *BnaA09g42390D*, *BnaA09g44370D*, and *BnaA09g44970D* were chosen as the candidates in yellow-seeded line No. 2127-17 [44]. Although four of them (*BnaA09.JAZ1*, *BnaA09.LOX3*, *BnaA09g44370D*, and *BnaA09g44970D*) were located within our QTL regions, only *BnaA09.LOX3* and *BnaA09g44370D* were down-regulated at 35 daf in our study, and none of them showed coexpressions with the flavonoid-related DEGs. We combined the QTL mapping with a differential expression analysis and WGCNA to identify seventeen genes as candidate genes. Among these, *novel4557*, *BnaA09G0616800ZS*, and *BnaC03G0060200ZS* were functionally associated with flavonoid biosynthesis and chosen as important candidates. However, further work is needed to confirm the genes responsible for the seed color formation in different yellow-seeded rapeseed varieties developed from different sources.

## 4. Materials and Methods

### 4.1. Plant Material and Growth Conditions

A set of 196 F_2_ individuals were derived from a cross between ZS11 (female parent producing black seeds) and HAZ (male parent producing yellow seeds). The resulting F_2_ plants were grown in the experimental field of the Hunan Agricultural University (Changsha, China), in accordance with conventional field cultivation (row spacing of 20 cm), and the parents were grown in plants in the experimental garden. Fresh leaf tissues of the parents and F_2_ individuals at the seeding stage were sampled for resequencing. Seed coats from the seeds collected from the main raceme and primary branches of the parents were sampled at 15, 25, and 35 daf for vanillin staining and a RNA-seq analysis. Three biological replicates were used for each experiment.

### 4.2. High-Throughput Sequencing and Genetic Linkage Map Construction

The genomic DNA of the parents and F_2_ plants, extracted via the cetyl-trimethylammonium bromide (CTAB) method, was subjected to WGR on the Illumina HiSeq PE150 platform, and paired-end reads that were 150 bp in length were obtained. Fastp (v0.23.0) [50] was used to trim the adapter sequences and reads of low quality (with parameters of: –W 4 –M 15), and reads shorter than 50 bp were also removed. The filtered clean reads were then mapped to the reference genome sequence of ZS11 [51] via the BWA (v0.7.15-r1140) [52], and GATK (v3.7) [53] was used to call the SNPs and InDels. A sliding window approach was used to construct a genetic bin map according to a published method [28]. Polymorphic markers between the two parental lines with aa × bb segregation patterns were retained and used to genotype the F_2_ individuals. High-quality SNPs and InDels were reserved using the following criteria: the minimum sequencing depth of each allele was 2 for the parents and 3 for the F_2_ plants; the minor allele frequency was larger than 30%; and the alleles were present in at least half of the F_2_ plants and fulfilled the ratio of marker segregation according to a chi-square test (*p* value ≥ 0.001). A window size of 15 SNPs or InDels was used for genotyping calling. Windows with 11 or more SNPs/InDels from either parent were considered to be homozygous for an individual, while those with fewer were considered to be heterozygous. Adjacent windows with the same genotype across the entire F_2_ population were merged into a recombination bin. Bins serving as genetic markers were employed for the construction of a genetic linkage map using MSTmap [54] and the genetic distances between these markers were calculated using the Kosambi mapping function.

### 4.3. Phenotype Evaluation and QTL Mapping

We used three methods (imaging, spectrophotometry, and visual scoring) to assess the seed color of the F_2_ individuals. The imaging method was conducted as follows: the seeds of each plant were divided into three groups (replications) and were then spread out with the hilum facing back to the scanning surface of a MicroTek scanner; the scanner was used to obtain images of the seeds, which were later used to analyze the seed color values (R, G, and B) with the phenoSEED software [55]. We also used a CM-2300D spectrophotometer from Konicaminolta to measure the seed color values (l, a, and b). The mean seed color value of the three replications was used as the phenotype value. Moreover, we classified the seeds into two grades using visual scoring: “0” was used for black or brown seeds, and “1” was used for those mixed with yellow seeds. As a result, we obtained seven seed color values to act as the phenotype values for the QTL mapping. The QTL analysis was performed using the Windows QTL Cartographer 2.5 software [56], using the composite interval mapping (CIM) model. The threshold logarithm of odds (LOD) score for each phenotype value was determined with a 1000-permutation test at a significance level of *p* < 0.05, and 1-LOD confidence intervals were used as the QTL confidence intervals.

### 4.4. Vanillin Staining and RNA-Seq

Vanillin staining can be used for the identification of seed color during early seed development [57]. Fresh seed coats were excised with pointed forceps, placed on a glass slide, and then incubated in a solution of 1.0% (*w*/*v*) vanillin in 5 N HCl for 20 min at room temperature. The samples were observed and imaged using a Nikon D5300 digital camera. The total RNA of the samples was extracted using an R6827 Plant RNA Kit (Omega). After the RNA quality (purity and integrity) was validated via agarose gel electrophoresis, a Nanodrop instrument, and a Qubit system, the resulting mRNA was purified using beads with Oligo (dT), and the cDNA was then synthesized. In total, 18 libraries were sequenced on the DNBSEQ-T7 platform. Adaptor sequences and low-quality reads were filtered using fastp (v 0.21.0) with default parameters.

### 4.5. Transcript Differential Expression and KEGG Enrichment Analysis

Clean reads of each sample were assembled and merged using StringTie (v2.1.4) [58]. The assembled transcripts were then compared to the transcripts of ZS11 [51] using gffcompare (v0.12.1) [59], with parameters of -R -C -K to identify new transcripts, which were translated into peptides using TransDecoder (v5.5.0) with the parameters: -m 50 -single_best_only (https://github.com/TransDecoder/). The clean reads were mapped to all the transcripts using Star (v2.7.9a) [60] with default parameters and the gene expression levels were measured using RSEM [61]. Genes for which the FPKM was <1 in all three biological replicates of the material were considered to be unexpressed. The DEGs were analyzezd using the DESeq2 (v1.26.0) [62] and screened with a threshold of |(FoldChange)| > 1 and false discovery rate (FDR) of < 0.05. KEGG annotations were performed using KOBAS (v3.0) [63] with a corrected *p* value < 0.05 based on the diamond BLASTP (v2.06.1) [64] results of all the protein sequences against proteins in the KEGG database. The KEGG pathway enrichment was conducted using clusterProfiler (v3.14.3) [65].

### 4.6. Whole-Genome Identification of Flavonoid-Related Genes and Validation of RNA-Seq by qRT-PCR

We used 56 flavonoid-related genes in *A. thaliana* to search for homologs in the ZS11 genome at BnPIR [66]. Newly assembled genes, whose protein sequences had best hits with the flavonoid-related genes in *A. thaliana* using BLASTP, were also included. The protein sequences of *A. thaliana* were downloaded from https://www.arabidopsis.org/. The integrative software TBtools (v1.108) [67] was used to illustrate the distribution of the flavonoid-related genes on the chromosomes of ZS11. DEGs were chosen for the validation of the RNA-seq data using qRT-PCR. Gene-specific primers were designed using Primer v3.0. All the genes, primers, and sizes of the amplicons are listed in (Appendix A). The subsamples for the RNA-Seq were first reverse transcribed into cDNA using an Evo M-MLV RT Mix Kit (Accurate Biotechnology, Changsha, China), according to the manufacturer’s protocol, and SYBRqPCR Master Mix (Vazyme) was used for a real-time qPCR on a Bio-Rad CFX-96 Real Time PCR System (Bio-Rad). The relative expression levels were analyzed using the –ΔΔCt method, with *BnaC02g00690D* (*ACT7*) used as an internal reference. Each sample included three biological replicates.

### 4.7. Construction of Gene Coexpression Networks and Prediction of Key Genes

We generated the scale-free coexpression networks based on the RPKM values of the expressed genes in the seed coats with the WGCNA package (v1.69) [68], with the parameters: Soft-Threshold = 8, minModuleSize = 100, and mergeCutHeight = 0.25. To identify the modules associated with the samples, we calculated the eigenvalue of each module and analyzed the Spearman correlations between the module eigenvalues and samples. In each of the coexpression modules, the KEGG enrichments were analyzed using clusterProfiler (v3.14.3) [65] and the pairwise coexpressed genes with weighted values of > 0.2 were retained.

The candidate genes for seed color were predicted based on a combination of QTL mapping, a differential expression analysis, and WGCNA. Firstly, the key stages and modules for the seed color formation were determined according to the vanillin staining, KEGG enrichment analysis, and WGCNA. Secondly, the genes within the QTL intervals that also showed differential expressions at the key stages were selected. Thirdly, the selected DEGs functionally associated with flavonoid biosynthesis or showing coexpression with the flavonoid-related DEGs in the key modules were selected as candidate genes. The coexpression network between the DEGs within the QTL intervals and the flavonoid-related DEGs was visualized using Cytoscape (v3.9.1) [69].

## 5. Conclusions

In this study, a high-density genetic map was constructed, which revealed a major QTL for seed color formation on chromosome A09 and a minor QTL on chromosome C03 in *B. napus*. The expression of flavonoid-related genes was down-regulated at 25 and 35 daf, which impeded the synthesis of flavonoids and contributed to the formation of yellow seeds. We further combined QTL mapping, a differentially expressed analysis, and coexpression network analysis, which resulted in the identification of seventeen candidate genes. Our results will facilitate the fine mapping of the responsible genes and the development of yellow-seeded *B. napus*.

## Figures and Tables

**Figure 1 ijms-24-09262-f001:**
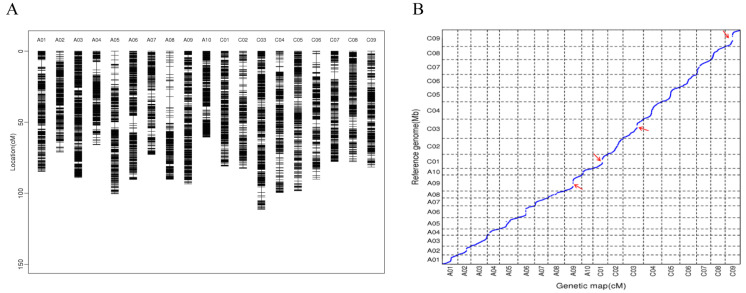
(**A**) Distribution of bin markers on the genetic map. (**B**) Collinearity analysis between the genetic map and the ZS11 genome, the red arrows indicate combination repression regions.

**Figure 2 ijms-24-09262-f002:**
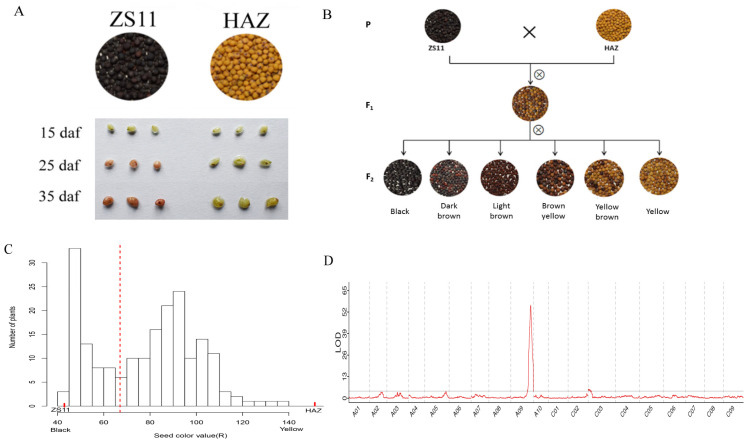
(**A**) Seed color differences and vanillin staining of seed coats of ZS11 and HAZ. (**B**) Seed color of parents, F_1_, and F_2_ plants. (**C**) Frequency distribution of seed color value (R) of F_2_ plants; and seed color values of parents are also showed (the red dashed line split the seeds into two grades on the basis of visual scoring). (**D**) Logarithm of odds (LOD) distributions on chromosomes according to seed color value (R).

**Figure 3 ijms-24-09262-f003:**
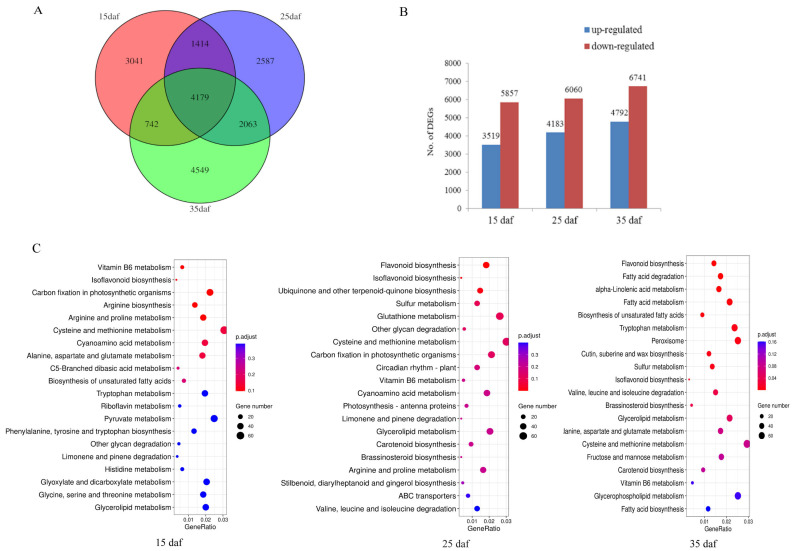
(**A**) Genes differentially expressed between yellow and black seed coats of rapeseed. (**B**) Number of up- and down-regulated differentially expressed genes. (**C**) The top 20 significantly enriched Kyoto Encyclopedia of Genes and Genomes (KEGG) pathways.

**Figure 4 ijms-24-09262-f004:**
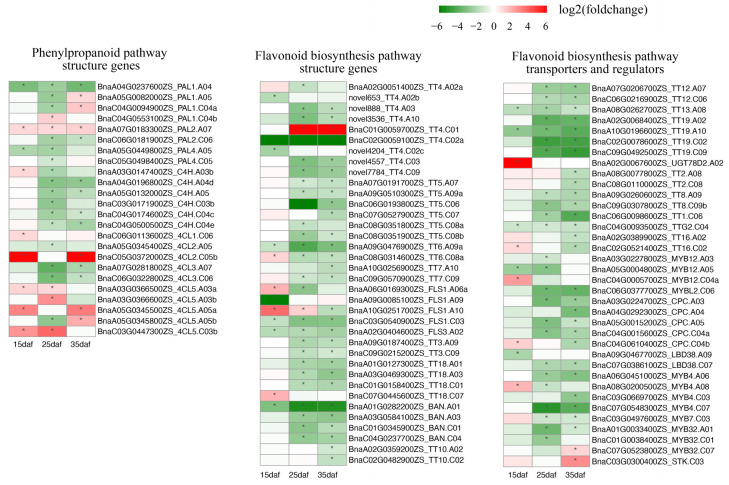
Heatmap for flavonoid-related differentially expressed genes during seed coat development; the red color indicates up-regulation, the green color indicates down-regulation, and the * in the box indicates significant difference.

**Figure 5 ijms-24-09262-f005:**
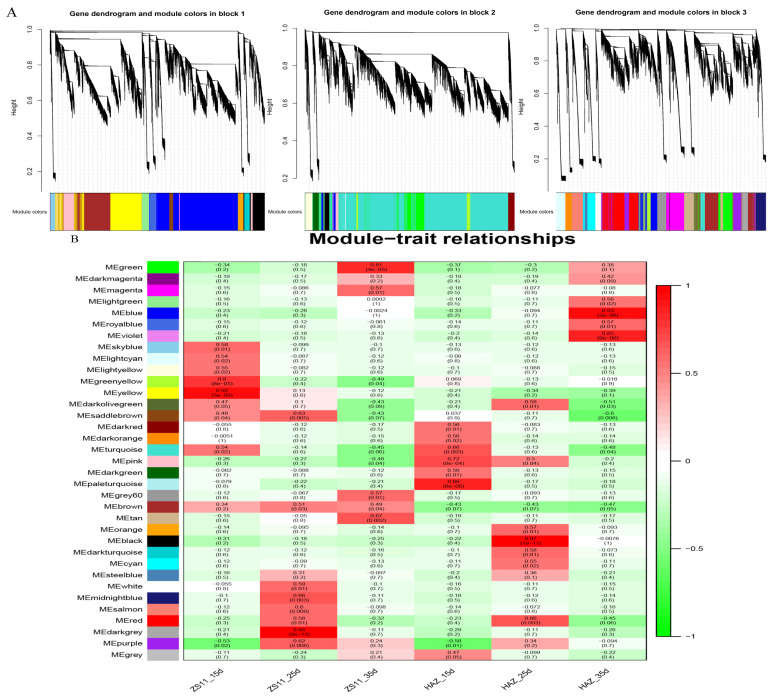
(**A**) Cluster of genes and construction of modules. (**B**) Heatmap of the correlations between modules and samples. Each row corresponds to a module and is labeled with a color as shown in the panel. (**C**) The top 20 significantly enriched Kyoto Encyclopedia of Genes and Genomes (KEGG) pathways for brown and darkgrey modules.

**Figure 6 ijms-24-09262-f006:**
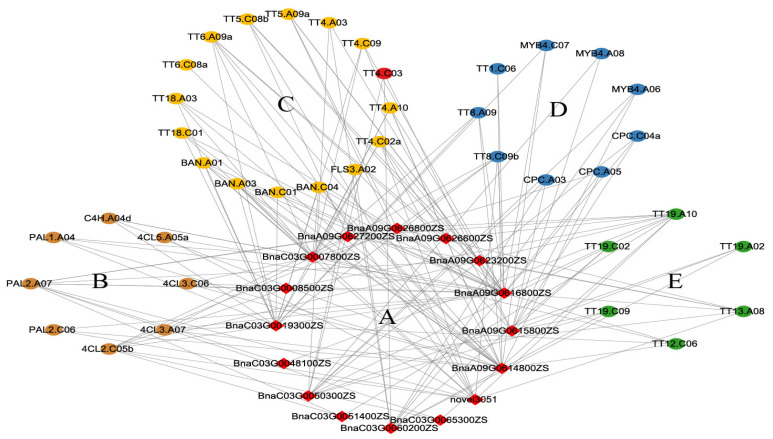
Coexpressed network between DEGs in QTL intervals and flavonoid-related DEGs, (**A**) is for DEGs in QTL interval. Flavonoid-related DEGs were classified into general phenylpropane pathway genes (**B**), structure genes (**C**), regulatory genes (**D**), and transporter-encoding genes (**E**).

**Table 1 ijms-24-09262-t001:** Summary of the genetic map.

Linkage Group	Length (cM)	No. Markers	Marker Interval (cM)	Max Interval (cM)
A01	84.693	226	0.376	5.068
A02	71.058	195	0.366	3.372
A03	89.030	295	0.303	2.042
A04	65.910	168	0.395	5.740
A05	100.404	228	0.442	3.937
A06	90.317	258	0.351	5.636
A07	72.777	180	0.407	2.400
A08	90.235	149	0.610	7.876
A09	93.434	276	0.340	1.787
A10	60.582	194	0.314	2.604
C01	80.946	236	0.344	1.531
C02	82.514	175	0.474	3.949
C03	111.315	303	0.369	3.891
C04	99.501	224	0.446	3.583
C05	98.286	241	0.410	2.604
C06	90.109	183	0.495	6.330
C07	77.757	207	0.377	5.120
C08	77.850	199	0.393	1.787
C09	81.611	211	0.389	2.298
Whole	1618.329	4148	0.400	3.766

**Table 2 ijms-24-09262-t002:** QTLs detected via different seed color values.

Trait	Chromosome	Position (cM)	LOD Value	R^2^ (%)	Confidence Interval (cM)	Physical Interval (bp)
R	A09	81.91	56.15	20.95	80.40–82.20	60,161,860–60,480,713
R	C03	0.31	5.52	6.25	0.00–1.00	296,168–3,736,656
G	A09	81.91	43.67	21.83	80.00–82.20	60,147,040–60,480,713
B	A09	81.91	47.41	19.61	80.30–82.10	60,147,040–60,480,713
B	C03	0.31	5.41	6.36	0.00–1.00	296,168–3,736,656
L	A09	81.91	31.92	15.63	79.90–82.20	60,113,334–60,480,713
a	A09	81.61	56.15	16.74	79.90–81.90	60,113,334–60,389,348
b	A09	81.11	39.96	10.91	79.90–82.20	60,113,334–60,480,713
b	C03	0.31	5.48	6.19	0.00–1.00	296,168–3,736,656
visual scoring	A09	84.81	70.15	21.19	84.50–85.10	60,954,990–61,041,243

QTL, quantitative trait locus; LOD, logarithm of odds; and R^2^, phenotypic variation explained.

## Data Availability

Transcriptome data used in this study are available in NCBI BioProject (PRJNA917831).

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
