# Peer review of "QTL Mapping and Transcriptome Analysis Reveal Candidate Genes Regulating Seed Color in Brassica napus"

_ijms, 2023, doi:10.3390/ijms24119262_

Round 1
Reviewer 1 Report
Dear authors,
I had reviewed the paper with title, QTL Mapping and Transcriptome Analysis Identifies Candidate Genes Controlling Seed Color in Brassica napus, which was submitted to the journal, International Journal of Molecular Sciences. Overall the design and quality of the study and results are qualified and ready for publishing. However, some revises are required before publication.
Minor revise:
1. The paper is written awkward and requires professional English editing.
2. Need full names for several abbreviation, for example, Chr, Pos, LOD, and R2 in Table 2, QTL, SNP, daf, etc. Please check the entire draft and provide the full names.
3. There should be a space between number and word/abbreviation. For example, the “35daf” and “~6Gb” on lane 151 should be “35 daf” and “~6 Gb”, “Figure2” on lane 143 and “Figure3” on lane 192 should be “Figure 2” and “Figure 3”. Please check the entire draft and correct all the errors.
The paper is written awkward and requires professional English editing.
Reviewer 2 Report
Oilseed rape, better known in binomial nomenclature as Brassica napus L. (Brassicaceae), is an ornamental vegetable ranked as the third largest oil crop worldwide. Its oil and proteins are used in food, livestock feed and industrial sectors. Several factors (e.g., seed colour, flowering time and light) have been reported to affect the seed size and oil yield of this species. Thus, various attempts are being made to optimize the conditions that gives the best yields from this oil crop. The submission by Liu et al. employed quantitative trait loci (QTL) mapping and transcriptome analysis for identifying genes responsible for regulating yellow color in Brassica napus seeds. The obtained results provide insights that could be useful for maximizing yields in this species.
I found the manuscript to be professionally written, but there are evident grammatical mistakes and incoherent abbreviations in the draft. Other than this, the manuscript could be accepted for publication after the following minor corrections.
1. Title & Abstract
L2, L14: QTL should preferably be expanded.
L9: HAZ >> (HAZ)
L23-25: Wasn’t this the aim of the current study? If not, which approach is required to confirm these genes?
L25: yellow seed colour?
2. Keywords
Words that appear in the title already are not useful as keywords. Please revise.
3. Introduction
L30: Brassica napus >> Brassica napus (Brassicaceae).
It would be nice to give a brief description of what this plant is before going into its uses. For example, it is also called Lewat, Oilseed rape, Reps or simply canola. It is a member of the cabbage (Brassica) genus within the cruciferous family.
Song et al. (2021). https://doi.org/10.3390/ijms22147559
L34: Brassica napus >> Brassica napus (B. napus). Thereafter, use the abbreviation B. napus throughout the draft.
L38: was >> is.
4. Results
L89: 196 F2 >> In total, 196 F2. It is rather confusing when numbers are used to start a sentence.
L: filtertion? Are the authors meaning filtering or filtration?
5. General Comments
-Does daf seen at lines (such as L118, L213, L361, L347 and L307) refer to ‘‘days after flowering’’. If so, (i) it should be expanded at first use in L118, (ii) this should be correctly written because some are written as 35daf while others appear as 35 daf.
-There are some considerable overlaps seen in some parts of the manuscript (see attached report).

-Various grammatical errors (e.g., past tenses instead of present continuous tenses are evident throughout the manuscript)
